# Mitigation of Galvanic Corrosion in Bolted Joint of AZ31B and Carbon Fiber-Reinforced Composite Using Polymer Insulation

**DOI:** 10.3390/ma14071670

**Published:** 2021-03-29

**Authors:** Jiheon Jun, Yong Chae Lim, Yuan Li, Charles David Warren, Zhili Feng

**Affiliations:** 1Materials Science and Technology Division, Oak Ridge National Laboratory, Oak Ridge, TN 37831, USA; liy5@ornl.gov (Y.L.); fengz@ornl.gov (Z.F.); 2Energy and Transportation Science Division, Oak Ridge National Laboratory, Oak Ridge, TN 37831, USA; cdavewarren@tds.net

**Keywords:** dissimilar material joint, carbon fiber reinforced composite, magnesium alloy, galvanic corrosion, mechanical joint integrity

## Abstract

The use of polymer insulation to mitigate galvanic corrosion was examined for bolted joints of AZ31B Mg alloy and carbon fiber-reinforced composite. To assess the corrosion behaviors of bolted joints with and without polymer insulation, solution immersion and salt spray exposure (ASTM B117) tests were conducted, and the corrosion depths and volumes were determined for the joint specimens after the tests. The polymer-insulated bolted joints exhibited much lower corrosion depths and volumes, highlighting the effective mitigation of galvanic corrosion. The reductions of joint strength in the post-corrosion joint specimens were relatively small (up to ~10%) in the polymer-insulated group but greater (up to 90%) in the group with no insulation. Cross-sectional characterization of post-corrosion joints with polymer insulation revealed local pits developed on AZ31B under galvanic influence, indicating that limited galvanic attack (that did not decrease the joining integrity significantly) could still occur during a long salt spray exposure (~1264 h) owing to the permeation of an aqueous corrosive medium.

## 1. Introduction

Multi-materials consisting of polymer composites (e.g., carbon fiber- or glass fiber-reinforced polymers) to lightweight metals (magnesium alloys, high-strength aluminum alloys, and advanced/ultra-high-strength steels) structures can effectively improve vehicle fuel efficiency to enable compliance with government regulations (i.e., for greenhouse gas emissions) [1]. Among lightweight metals, magnesium (Mg) alloys are the lightest structural materials [2] and can lead to weight reductions of up to 55% when used instead of conventional steels [1]. Highly engineered polymer composites such as carbon fiber-reinforced composites (CFRC) are another low-density material option that provides superior mechanical strength, corrosion resistance, and design flexibility [3]. Conceptually, noble hybrid autobody structures composed of Mg alloys and CFRC can result in significant weight reduction for improved fuel efficiency. However, joining of lightweight metals to polymer composites is one of the key technical obstacles in such multi-material autobody structures.

Extensive research and development efforts have been made to advance multi-material joining techniques, including fusion welding [4,5], ultrasonic welding [6], friction-based joining [7,8,9,10,11,12], mechanical fastening [13,14,15], adhesive bonding [11,16] and weld bonding (use of adhesive combined with other joining techniques) [11,17]. However, these techniques are not fully developed to ensure a reliable joining of Mg alloy and CFRC. For example, fusion welding, ultrasonic welding, and some friction-based joining techniques are only used to join thermoplastic polymer to metals joints, while the joining of thermoset polymer composites to metals is not feasible. Conventional mechanical fastening such as self-piercing riveting, may induce cracking in low ductility materials such as Mg alloys at room temperature [18]. Although adhesive bonding has been widely used for automotive and aerospace applications with good bonding strength, notable drawbacks for overall joining reliability include low peel strength, poor impact performance, and environment degradation.

Bolting is one of the most well-established joining techniques and is widely used in transportation sectors because it provides for relatively easy assembly of similar or dissimilar materials, as well as and disassembly of joints for repair and recycle. Bolting also provides good mechanical joint performance [13]. However, bolting requires a pre-drilled hole to assemble the materials, which can lead to damage on polymer composites [19]. To minimize the mechanical damage from hole drilling, selection of an optimum machining technique is critical.

Another major concern in any dissimilar material joint with Mg alloys is galvanic attack, in which Mg suffers from accelerated corrosion [12]. This occurs because Mg is one of the most anodically active metals and will readily dissolve to form metal ions. To mitigate galvanic corrosion, electrical insulation can be placed between dissimilar materials to increase resistance at the contact interface of anodic and cathodic materials. For a bolted joint of AZ31B Mg alloy and CFRC, electrical insulation can be applied at the contact interfaces of Mg with a steel bolt or Mg with a carbon fiber bundle of CFRC to mitigate galvanic corrosion. In this work, polymeric materials, including curable epoxy resins and polytetrafluoroethylene (PTFE), were selected to electrically insulate AZ31B and other components of the bolted joints.

To perform a quantitative evaluation of Mg galvanic corrosion in dissimilar material joints, a method was implemented to selectively expose Mg anode and coupled cathode surfaces by masking other areas of the joint with polymer tapes [12]. The selectively exposed Mg anode developed a corrosion volume which allowed quantitative evaluation and visualization of galvanic attack in a dissimilar joint. This research also adopted a similar selective corrosion exposure method for the bolted joints, along with the conventional salt spray exposure test that has previously been used by others for dissimilar material joints [20,21,22,23].

In the present work, bolted joints of AZ31B Mg alloy and CFRC were prepared with and without polymer resin with and without PTFE tape insulation, and the joint specimens underwent corrosion testing by 0.1 M NaCl solution immersion and ASTM B117 salt spray exposure. The corrosion volumes that developed on the exposed Mg areas were then compared for the polymer-insulated and bare (i.e., without polymer insulation) joints to evaluate the effectiveness of polymer-based insulation on galvanic corrosion mitigation. The strength of bare and polymer-insulated joints was also measured before and after the salt spray exposure tests. Finally, cross-sectional characterization of polymer-insulated joints before and after corrosion was conducted to investigate the corrosion attack that occurred at the joints’ inner dissimilar material interfaces.

## 2. Materials and Methods

### 2.1. Materials

A commercially available 3.29-mm-thick AZ31B Mg alloy was used for the top sheet material.

Table 1 summarizes the chemical composition of AZ31B from the manufacturer’s report. For the bottom sheet, a 4 mm thick thermoset carbon fiber reinforced composite was fabricated with the G-83 prepreg (T700, Toray) in a +45°/−45° stacking sequence with 20-ply layup (Clearwater Composites, Duluth, MN, USA). The mechanical properties of AZ31B and CFRC used in this work are summarized in Table 2.

Both AZ31B and CFRC were machined by waterjet cutting for the assembled parts of lap and cross-tension type joints. The appearance and dimensions of machined AZ31B and CFRC parts are shown in Figure 1a,b. For bolting, commercially available mechanical fasteners, steel bolts and nuts with yellow zinc plating finish, and type 316 stainless steel (ss316) washers were purchased from an external vendor (Mcmaster carr, Elmhurst, IL, USA). A bolt and a nut with two ss316 washers, which were used in this work, are shown in Figure 1c. Two commercial epoxy resins were purchased to prepare the polymer-insulated bolt joints: (1) a clear adhesive type resin (Master Bond EP33CLV) for coating on the AZ31B surface, and (2) a low-viscosity type resin (DAP fast cure epoxy-acrylate) for (a) healing of microcracks that would potentially be produced during a predrilled hole process and (b) for electrical insulation around a pilot hole on CFRC. A thin film PTFE tape was also purchased for the threads on bolts.

### 2.2. Assembly of Bolted CFRC-AZ31B Joints

The lap and cross-tension bolted joints are schematically illustrated in Figure 2a. Cross-sectional views of bare bolts, without polymer insulation, and polymer-insulated bolt joints are depicted in Figure 2b,c, respectively. The most probable galvanic paths (red and green dot arrows) in a bare bolted joint are described in Figure 2b. The primary purpose of polymer insulation is to prevent/inhibit the formation of such galvanic paths, as shown in Figure 2c. For the assembly of polymer-insulated joints, the following treatments were conducted for each part: (1) an adhesive-type epoxy was painted onto the top and bottom AZ31B surfaces around the pilot hole, (2) a low-viscosity epoxy was applied on the pilot hole surfaces of CFRC to fill in the exposed microcracks formed in the composite during the hole drilling, and (3) a thin PTFE tape was wrapped around the threaded sections of the steel bolts. As illustrated in Figure 2c, the combination of the polymer resin and PTFE tape application is designed for electrical insulation between AZ31B, CFRC, and the bolt setup. To avoid any potential damage on precoated epoxy resin on the AZ31B surface, a moderate clamping torque of 5.65 N·mm was applied to tighten the mechanical fasteners (i.e., bolt and nuts).

### 2.3. Corrosion Testing

Solution immersion tests were performed for the polymer tape-masked cross-tension joints illustrated in Figure 3a. These masked joint samples were partially immersed in 0.1 M NaCl solution, as depicted in Figure 3b, to allow corrosion of AZ31B surfaces at the bottom and inside the joint gap section. With this specific immersion test, the degree of Mg corrosion can be visually quantified by analyzing the corrosion volume formed after the immersion test. A photo of solution immersion test and another photo of a post-immersion bolt joint that formed corrosion volume at the bottom are shown in Appendix A. Several tape-masked cross-tension specimens—bare and polymer-insulated—were used with increasing solution immersion time. For one selected bare sample and one selected polymer-insulated sample, a reference saturated calomel electrode (SCE) was placed between the bolt head and the AZ31B bottom surface to measure the corrosion potential at the end of the immersion test.

A salt spray corrosion test according to ASTM B117 was also conducted for several bare and polymer-insulated joints with increasing exposure time. Both lap and cross-tension type joints were used, with tape masking applied in the end section(s) of AZ31B to expose only the central section of joints where galvanic corrosion would be most significant. The exposed areas of AZ31B in the central sections of the lap and cross-tension joints were 38 × 43 mm^2^ and 50 × 60 mm^2^, respectively, excluding the bolted centers. The photos of bolted joints with tape masking for salt spray tests are shown in Appendix A. The tape-masked joints were then loaded onto the plastic racks in a salt spray chamber, which is also shown in Appendix A. The salt spray corrosion test experienced several off times during which no salt spray was generated due to an instrumental limitation. These off times mostly lasted for 48 to 72 h during weekends, but longer in certain circumstances. The off times were not considered as part of the total exposure time. Another note is that tap water was used for pressurized steam and 3.5 wt.% NaCl solution rather than 4 Mohm–grade distilled water. The tap water exhibited 3.7 × 10^−3^ Mohm and pH 7.9, and it contained some ionic species at relatively low concentrations, as presented in Appendix A.

To remove corrosion products without using chemical agents, all post-corrosion joint samples were cleaned by ultrasonication in distilled water and mechanical rubbing with hard polymer brushes. This cleaning method was repeated at least three times until no further removal of corrosion products was observed. Overall, this physical cleaning successfully removed most visible corrosion products from the joint specimens.

### 2.4. Static Lap Shear Tensile Testing

The mechanical joint integrity before and after the salt spray corrosion test was evaluated by static lap shear tensile testing using an MTS tensile machine with a constant crosshead speed of 10 mm·min^−1^ at room temperature. To avoid a bending effect during tensile shear testing, spacers were used to clamp the lap shear coupons to align them vertically between the grips.

### 2.5. Characterizations (Optical and Electron Microscopy)

To quantify the corrosion volume after the salt spray exposure test, an optical profilometry instrument (Keyence, Osaka, Japan) was used. The baseline and post-corrosion samples were mounted in epoxy and cut into halves using a diamond saw to obtain the cross sections. The cross-sectioned samples were then ground using silicon carbide (SiC) papers with 600, 800 and 1200 grits, followed by fine-polishing with diamond progressively finer suspensions 3, 1, and 0.5 µm using a Struers auto-polisher machine. A Zeiss Axio microscope was used for optical characterization. The images were stitched using the MosaiX mode. Scanning electron microscopy (SEM) and energy dispersive X-ray spectroscopy (EDS) characterizations were carried out using TESCAN MIRA3 with an acceleration voltage of 20 kV.

## 3. Results

### 3.1. 0.1 M NaCl Immersion Test

The lower center sections of AZ31B after 0.1 M NaCl immersion tests are shown in Figure 4 for the bare and polymer-insulated cross-tension joints. The bare joints underwent up to 314 h immersion (Figure 4a), and the material loss by galvanic corrosion formed relatively uniform depths from the initial exposed area (bottom surface, see Figure 3). For the longer immersion times (Figure 4b), the bare joints showed greater corrosion depths, whereas the polymer-insulated joints only formed “crevices”, without uniform material loss. The corrosion potential measured after 614 h immersion were −1.26 and −1.48 V_SCE_ for the bare and polymer-insulated joints, respectively, indicating a higher anodic polarization in the bare case due to greater galvanic impact. Note that AZ31B in 0.1 M NaCl at room temperature commonly exhibits −1.55 ~ −1.6 V_SCE_, with no galvanic coupling [12,24,25]. Apparently, the polymer-insulation method was effective in mitigating galvanic corrosion of AZ31B in the bolted joint configuration used in this work.

For a quick quantification of Mg galvanic corrosion, the depths of corroded volumes were measured for AZ31B in the bare and polymer-insulated bolted joints, as designated by yellow arrows in Figure 4b. The measured corrosion depths are plotted as a function of immersion time in Figure 5. The depth formed from the bare joints increased with time but seemingly at a slower rate after 314 h. This slow-down is likely associated with the lateral-direction corrosion loss, which was noticed in the samples immersed for 314, 404 and 623 h (see Figure 4). Meanwhile, there was only one polymer-insulated joint sample that developed a measurable corrosion depth after 623 h. The corrosion depth was about 1.7 mm and was much smaller than the depth measured from the bare joints (≥5 mm). In light of these results, together with the corrosion potential data, it can be stated with confidence that Mg galvanic corrosion was mitigated by adopting a polymer insulation method.

### 3.2. ASTM B117 Salt Spray Exposure

The bare and polymer-insulated joints were also exposed to an ASTM B117 salt spray environment with the specific masking applied on both lap and cross-section configurations, as described in the experimental section. The post-corrosion lap sheer joints are shown in Figure 6 for the bare and polymer-insulated conditions. The bare joints developed corrosion ditches on Mg (designated as p1-3 in the top row of Figure 6) around the ss316 washers, clearly indicating severe Mg galvanic corrosion proximity to the noble metallic components of the bolting setup (which were galvanically protected, and therefore not corroded). It was visually noted that the corrosion ditch fully penetrated the AZ31B sheet after 438 h exposure. In polymer-insulated joints exposed longer than 438 h, however, no distinct groove attack was found around the ss316 washers, and both the steel bolts and AZ31B corroded in a relatively uniform manner. This observation indicates that the polymer insulation was effective in impeding galvanic coupling so that AZ31B did not suffer from severe corrosion ditch attack, whereas the steel bolt, which was no longer galvanically protected, corroded simultaneously.

Figure 7 shows the cross-tension joints after salt spray exposure for one bare case (238 h) and three polymer-insulated cases (238, 438, 732 h). Again, the formation of a severe corrosion ditch with an uncorroded steel bolt was observed in the bare joint. In the polymer-insulated joints, no corrosion ditch was visually detected, and the steel bolts formed rust layers that covered an increasingly large area over time. Overall, the corrosion attacks and morphologies observed after the salt spray exposure were very similar between the lap and cross-tension joints prepared with the bare and polymer-insulated conditions.

The corrosion volumes of the bare and polymer-insulated joints were quantitatively analyzed using an optical profilometry technique, with the uncorroded (i.e., tape-masked during the salt spray exposure) AZ31B surface serving as the reference plane. The post-corrosion joints were disassembled, and only the AZ31B plates were used for optical profilometry measurements, as described further in Appendix A. The quantified corrosion volumes are plotted as a function of salt spray exposure time in Figure 8. In both the lap and cross-tension joints, the bare condition showed the corrosion volumes increasing with time, with a much greater increase than seen in the polymer-insulated condition due to the prevailing and aggregating corrosion ditch attack. In contrast, the corrosion volumes of polymer-insulated joints were relatively low and did not increase with time, implying that Mg corrosion was saturated without being alleviated by the galvanic impact that was prominent in the bare joints.

### 3.3. Mehanial Joint Performance and Fractograph

Figure 9a,b depicts representative load and displacement curves from lap shear testing for bare and polymer-insulated bolted CFRC-AZ31B joints with different corrosion exposure time. The bare joints showed greater reduction of failure load and elongation at failure with increasing exposure time. In contrast, the failure loads and displacements for polymer-insulated joints were not decreased significantly in any of the cases, even after 1264 h exposure. Figure 9c,d present the averaged lap shear peak failure load for the bare and polymer-insulated bolted joints with increasing ASTM B117 salt spray exposure times. For the bare condition, the averaged failure load for uncorroded joints was 15.96 ± 0.32 kN. This is higher or comparable to the metal-to-composite joints fabricated by other technologies and summarized in the open literature [11,26]. As corrosion exposure time increased, the averaged peak failure load was greatly decreased, as shown in Figure 9a. Finally, the retained failure load was only 1.64 ± 0.33 kN at 438 h due to significant galvanic corrosion on AZ31B. Figure 9b shows the averaged peak shear failure load for the polymer-insulated bolted joints at up to 1264 h corrosion exposure time. The averaged failure load for uncorroded joints was 15.51 ± 0.07 kN, which is similar to the value of the bare uncorroded joints. Although some deviations of the average failure loads were observed, 80~90% of the original joint failure load was generally obtained in the polymer-insulated joints after the salt spray exposures.

Table 3 and Table 4 summarize the averaged lap shear failure load, elongation at failure, and retained joint strength for both conditions at different corrosion exposure times. In the comparison at 438 h of exposure time, only 10.3% of the original strength was retained for the bare joints, whereas 78% of the original joint strength was retained for the polymer-insulated joints. The average elongation for the bare joints significantly reduced as the corrosion exposure time increased due to the galvanic corrosion of Mg alloy in the joint. However, the polymer-insulated joints showed relatively small reduction of the averaged elongation at failure because galvanic corrosion of AZ31B was mitigated by electrical insulation between the metal interfaces. Overall, the polymer-insulation method used in this work was effective in reducing galvanic corrosion of Mg alloy, thereby mitigating the degradation of joint integrity.

Figure 10 shows fractography for the bare (Figure 10a–d) and polymer-insulated bolted joints (Figure 10e–h) with increasing corrosion exposure times. For the bare case, the failure mode for uncorroded joints was mixed cleavage-tension (red arrow #1) and net-tension (red arrow #2) failures, as shown in Figure 10a. As corrosion exposure time increased, galvanic corrosion of AZ31B formed a corrosion ditch on the periphery of the bolted center, as previously shown in Figure 6. This ditch corrosion attack on AZ31B resulted in the crack initiation, as designated by red arrows in Figure 10b,c, and it subsequently led to final failure during the tensile shear testing. For the polymer-insulated joints, mixed cleavage tension and net tension failure modes were commonly found in uncorroded and salt spray-exposed conditions, indicating that the primary failure mode was not changed due to the mitigation of AZ31B galvanic corrosion.

### 3.4. Cross-Sectional Characterization

The cross-sectioned lap joints with polymer insulation are compared for uncorroded and corroded (by salt spray exposure for 1264 h) conditions in Figure 11. The comparison of low-magnification optical images (Figure 11a,b) reveals corrosion pits formed in AZ31B around the ss316 washer (denoted as pitting), and it also shows corrosion loss of AZ31B underneath the washer and bolt (A and B vs. C, D and E). The corrosion pits near the washer that were not identified by visual inspection indicate that the galvanic effect was mitigated but was not completely removed by applying polymer insulation on the AZ31B surface. A notable corrosion loss of steel bolt (F) is also observed at a contact point with the ss316 washer, indicating possible local galvanic corrosion of steel by ss316.

A gap between the washer and AZ31B of the uncorroded joint (A in Figure 11a) is shown at high magnification in Figure 11c, revealing that the pre-coated, porous, upto 200 µm thick epoxy layer adhered firmly on AZ31B. A similar gap location of a corroded joint (C in Figure 11b) is presented in Figure 11d that exhibited shallow and deep pits on AZ31B. This observation indicates that the pre-coated epoxy layer was not fully protective, presumably due to the permeation of the aqueous corrosive medium. In the corroded joint, an EDS mapping analysis was conducted for the area designated by C and D in Figure 11b, as presented in Figure 11e. The overlap of O and the relatively low Mg signals indicate the corroded area in AZ31B. It is also noted that the ss316 washer exhibited strong Fe without any overlap with O, indicating that the washer remained uncorroded.

More EDS mapping analyses were conducted for the areas designated by B and E in Figure 11a,b, respectively, as presented in Figure 12. In the uncorroded joint, Fe and Mg did not overlap with O, and a strong F intensity was detected from the PTFE tape (Figure 12a). In contrast, extensive corrosion of AZ31B was indicated by the overlap of Mg and O, as well as corrosion of steel bolt threads (overlap of Fe and O) underneath the PTFE tape as designated by F intensity) (Figure 12b). Despite the corrosion mitigation effect, the application of polymer insulation could not completely stop corrosion of AZ13B and the steel bolt in the inner area of the joint.

## 4. Discussion

The corrosion tests performed in this work—NaCl solution immersion and salt spray exposure—clearly showed the mitigation of galvanic corrosion attack by implementing a polymer-insulation method. However, galvanic corrosion attack at the local spots was still observed in a polymer-insulated joint specimen with a relatively long salt spray exposure (1264 h), implying that the polymer-insulation method cannot completely prevent the formation of galvanic coupling. It is considered that the main cathodic surface(s) for galvanic corrosion are the steel bolt head and ss316 washer for the bare joints, but only the ss316 washer for the polymer-insulated joints. This is because the steel bolts were galvanically protected in the bare joints but not in the polymer-insulated joints. To improve the protection against galvanic corrosion, an insulation treatment on the ss316 washer should be implemented so that the washer no longer functions as a cathode in a polymer-insulated joint. Future work can focus on insulation treatment(s) for the washer and verification of galvanic corrosion prevention in a bolted joint configuration.

Regarding mechanical joint integrity, the polymer-insulation method was effective in minimizing joint strength reduction by preventing corrosion ditches that induced crack initiation and propagation under tensile loading. The polymer-insulated joints in uncorroded and corroded conditions shared the same primary failure mode, but the joint strength was reduced by up to 20% after corrosion. One potential cause of this reduction is presumably associated with the local corrosion pits found around the washer (see Figure 11b); the corrosion pits in the path of the shared failure mode likely facilitated crack propagation to the final failure. Meanwhile, the reduction of joint strength in the bare bolted joint of AZ31B and CFRC was very significant after a relatively short salt spray exposure (≤438 h) due to the formation of corrosion diches. In contrast, another joint configuration in which carbon fiber epoxy composites were rivet-joined to Al alloy 6060 showed much lower joint degradation by retaining 77% of the original strength after salt spray exposure for 1176 h [23]. This highlights the greater susceptibility of Mg alloys to galvanic corrosion and subsequent mechanical degradation when joined with a noble metal fastener. To avoid any early corrosion failure of the joints, it is also crucial to determine effective corrosion barrier(s) for Mg alloys. The results of the current work can be applied to the joint design for galvanic corrosion mitigation in lightweight multi-material vehicles and other transportation industries requiring fuel-efficient improvement.

## 5. Conclusions

In this work, polymer insulation of bolted AZ31B Mg alloy and CFRC joints was applied to remove the galvanic circuit formation which would cause accelerated anodic dissolution of Mg by cathodic hydrogen reduction on a steel bolt and a nut, ss316 washers, and, to lesser degree, carbon fiber bundles in the composite. The joint specimens were prepared in lap and cross-tension joint configurations in bare and polymer-insulated conditions. After application of different tape masking, the joint specimens underwent corrosion tests by immersion in 0.1 M NaCl and salt spray exposure. The results are summarized below.
The corrosion depths of AZ31B measured after the immersion tests were much greater in the bare (i.e., no insulation as the control case) joints than in the polymer-insulated bolted joints, indicating that polymer insulation applied on bolted joints effectively reduced galvanic corrosion.After the salt spray exposure tests, the bare joint developed corrosion ditches around the washers, whereas the polymer-insulated joints did not have any severe attack in the same location. The corrosion volume determined by optical profilometry was greater in the bare joints than in the polymer-insulated joints.Only about 10% of joint strength remained in the bare joints after 438 h salt spray exposure, with the failure initiated at a corrosion ditch of the AZ31B surface. In contrast, 80~90% of joint strength remained in the polymer-insulated joints after 1264 h in the failure mode, a strength similar to the uncorroded specimens.Post-corrosion polymer-insulated joints (1264 h salt spray exposure) revealed local corrosion pits on the surface of the AZ31B adjacent to the washer, as seen in a cross-sectional characterization, indicating that polymer insulation did not completely remove the galvanic corrosion, but it functioned as a mitigation method.

## Figures and Tables

**Figure 1 materials-14-01670-f001:**
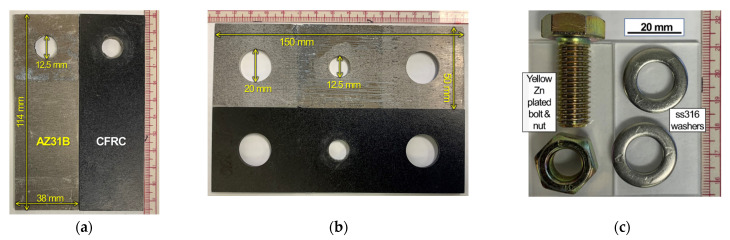
Machined AZ31B and CFRC sheets with 12.5 mm pilot holes for (**a**) lap shear and (**b**) cross-tension joint specimens. For bolting in a pilot hole, a steel bolt and a nut with yellow Zn plating, as well as two ss316 washers shown in (**c**) were used.

**Figure 2 materials-14-01670-f002:**
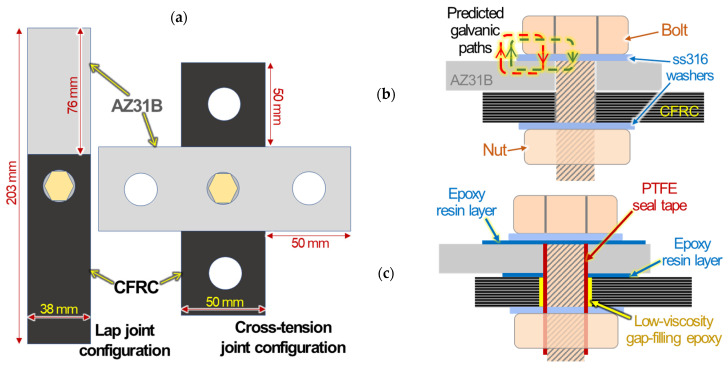
Schematics of (**a**) lap and cross-tension joints of AZ31B and CFRC. The cross sections of bolted AZ31B and CFRC are also schematically described for (**b**) bare joint with predicted galvanic paths (red and green dot arrows) and (**c**) polymer-insulated joints.

**Figure 3 materials-14-01670-f003:**
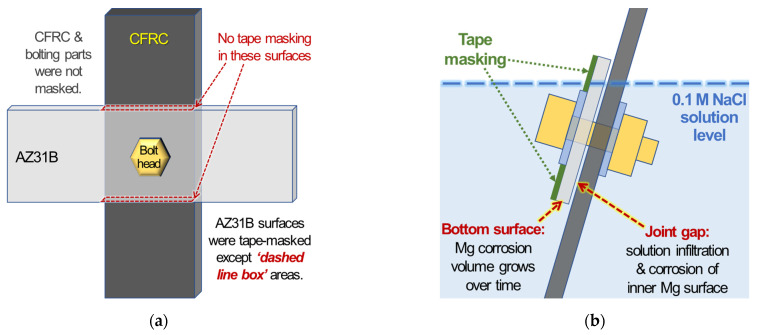
Schematics of (**a**) exposed AZ31B surfaces without tape masking in a cross-tension joint and (**b**) an immersed joint specimen with tape masking from the side view.

**Figure 4 materials-14-01670-f004:**
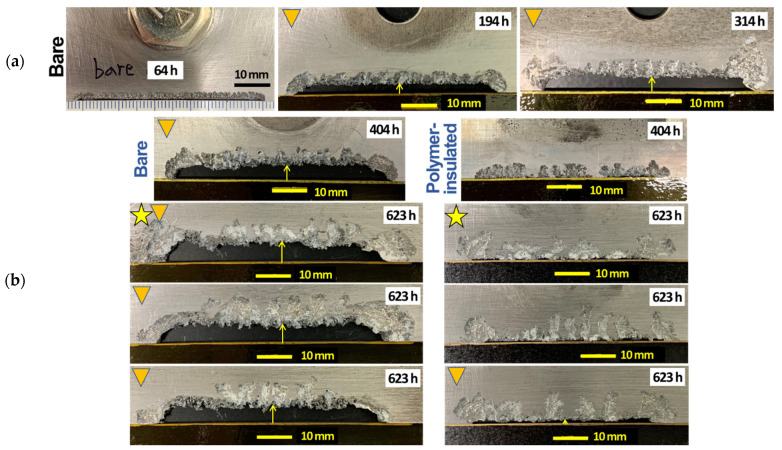
Post-immersion AZ31B sheets that developed corrosion volumes from (**a**) bare bolt joints with immersion time up to 314 h, and (**b**) bare and polymer-insulated joints with immersion times of 404 and 623 h. The triangle symbols indicate corrosion depth measurement, and star symbols indicate corrosion potential measurements in which the values at the completion of the immersion tests were −1.26 and −1.48 V_SCE_ for the bare and polymer-insulated joint specimens, respectively.

**Figure 5 materials-14-01670-f005:**
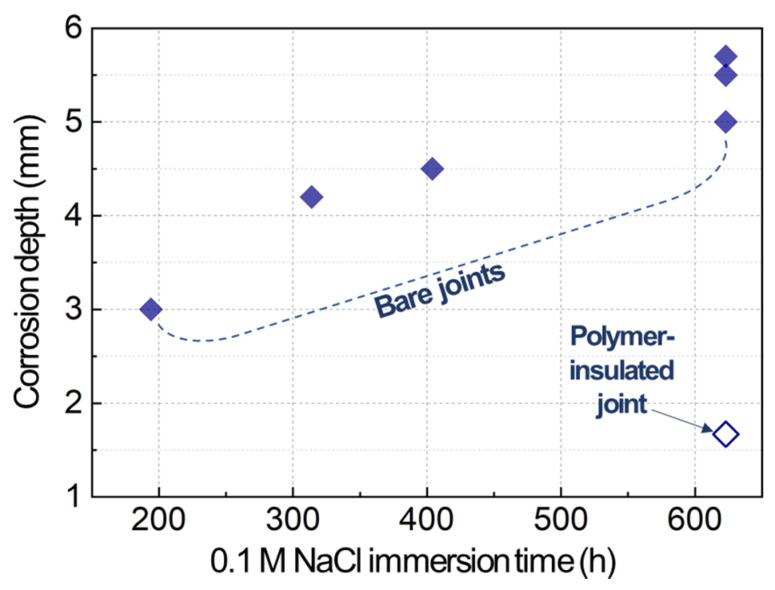
Corrosion depths measured from post-immersion AZ31B shown in Figure 4 as a function of immersion time.

**Figure 6 materials-14-01670-f006:**
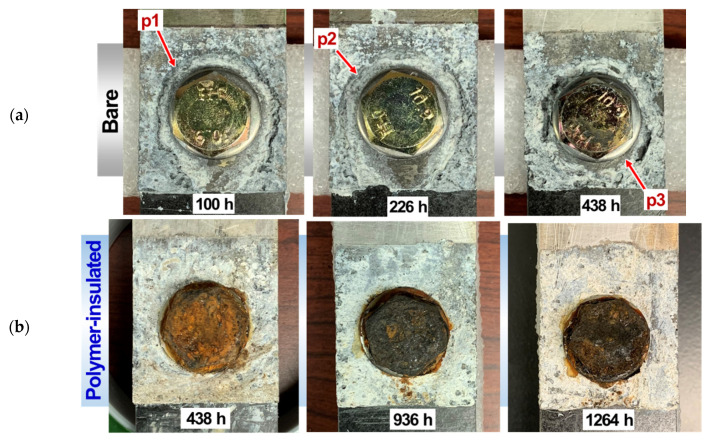
Lap shear bolted joints after salt spray exposure presented for (**a**) bare and (**b**) polymer-insulated conditions. The exposure time is designated for each specimen.

**Figure 7 materials-14-01670-f007:**
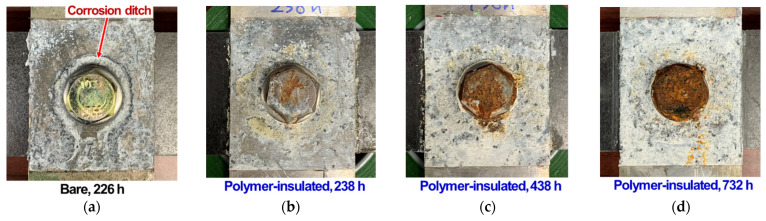
Cross-tension bolted joints after salt spray exposure presented for (**a**) bare and (**b**–**d**) polymer-insulated conditions. The exposure time is designated for each specimen.

**Figure 8 materials-14-01670-f008:**
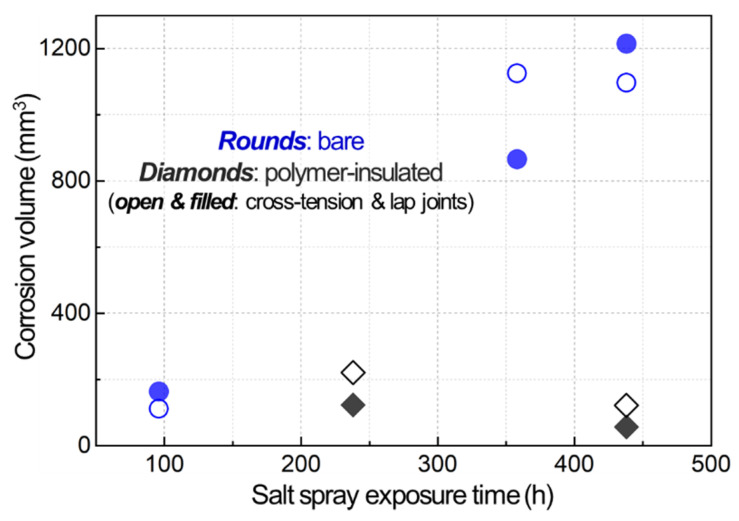
Corrosion volumes of lap and cross-tension joints plotted as a function of salt spray exposure time for the bare and polymer-insulated conditions. The corrosion volumes were determined by optical profilometry, as exemplified in Appendix A.

**Figure 9 materials-14-01670-f009:**
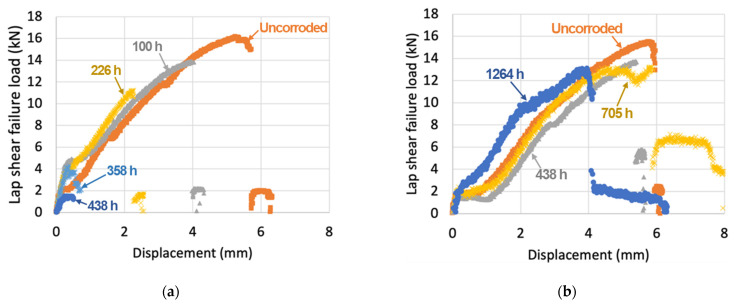
Representative load and displacement curves from lap shear tensile testing for (**a**) bare and (**b**) polymer-insulated bolted CFRC-AZ31B joints with different ASTM B117 exposure times. The average values of lap shear peak failure load are presented for (**c**) bare and (**d**) polymer-insulated bolted joints with the different exposure times.

**Figure 10 materials-14-01670-f010:**
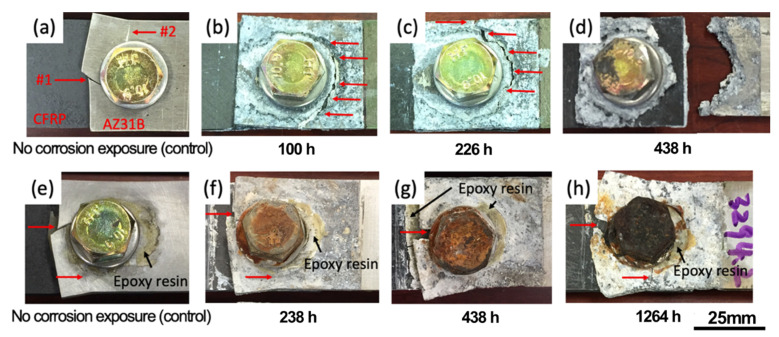
Comparison of fractography from lap shear tensile testing with different ASTM B117 exposure times for (**a**–**d**) bare and (**e**–**h**) polymer-insulated CFRC-AZ31B bolted joints. Red arrows indicate the failure locations on AZ31B, and black arrows show the epoxy resin overfill layer on the AZ31B surface.

**Figure 11 materials-14-01670-f011:**
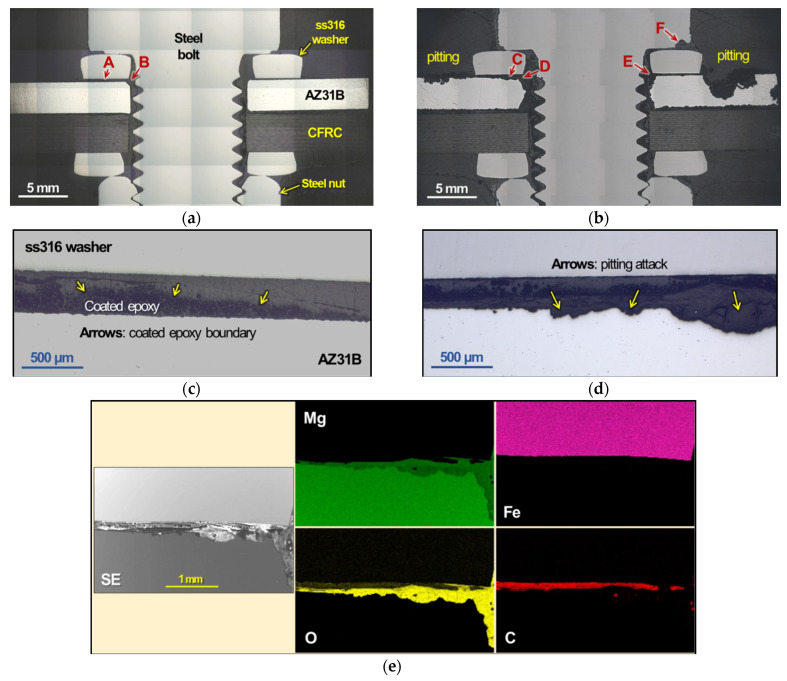
Light microscopy images of cross-sectioned polymer-insulated bolted joints (**a**,**c**) in as-joined condition and (**b**,**d**) after salt spray exposure for 1264 h. The local areas designated *A* in (**a**) and *C* in (**b**) are shown in a higher magnification in (**c**,**d**), respectively. An SEM image is presented in (**e**), and an EDS mapping analysis for a local area designated by *C* and *D* is presented in (**b**). *SE* indicates the secondary electron image mode. All EDS elemental maps were recorded using K-alpha characteristic X-ray of each element.

**Figure 12 materials-14-01670-f012:**
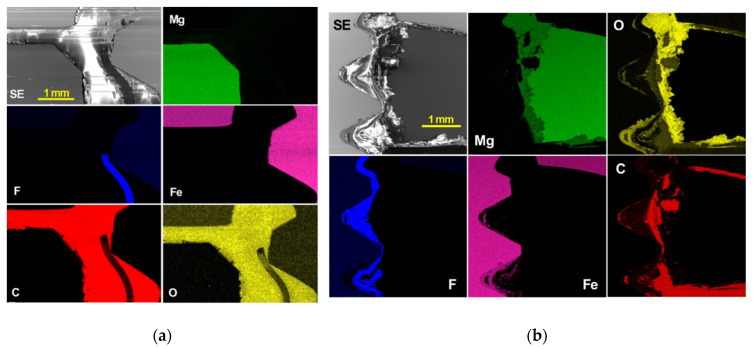
SEM images with corresponding EDS maps for (**a**) the area designated by B in Figure 11a and (**b**) the area designated by E in Figure 11b. All EDS elemental maps were recorded using K-alpha characteristic X-ray of each element.

**Table 1 materials-14-01670-t001:** Summary of chemical compositions for AZ31B from material specification sheet.

Element	Al	Cu	Mn	Zn	Ca	Ni	Be	Si	Fe	Other
Wt%	3.03	0.001	0.42	1.08	0.001	0.001	<0.001	0.015	0.0025	<0.3

**Table 2 materials-14-01670-t002:** Summary of mechanical properties for AZ31B and CFRC (for CFRC longitudinal, carbon fibers are aligned in 45° angle to the tensile direction, and for CFRC 45°, carbon fibers are parallel to the tensile direction).

Properties	Yield Strength (MPa)	Ultimate Tensile Strength (MPa)	Elongation (%)
AZ31B	220.5	299.5	13.05
CFRC (Longitudinal direction)	96.3	194	22.7
CFRC (45° direction)	N/A	907.7	0.23

**Table 3 materials-14-01670-t003:** Summary of lap shear tensile testing for the as-bolted CFRC-AZ31B joints.

ASTM B117 Exposure Time (h)	Peak Fracture Load (kN)	Elongation at Failure (mm)	Retained Strength (%)
0	15.96 ± 0.32	7.16 ± 1.25	100
100	13.85 ± 1.81	4.84 ± 1.20	86.8
226	11.59 ± 1.93	3.58 ± 1.15	72.7
358	5.18 ± 1.19	2.02 ± 1.11	32.5
438	1.64 ± 0.33	0.45 ± 0.34	10.3

**Table 4 materials-14-01670-t004:** Summary of lap shear tensile testing for three-step insulated bolted CFRC-AZ31B joints.

ASTM B117 Exposure Time (h)	Peak Fracture Load (kN)	Elongation at Failure (mm)	Retained Strength (%)
0	15.51 ± 0.07	6.45 ± 0.31	100
238	13.55 ± 0.53	7.42 ± 0.59	88.3
438	12.09 ± 1.48	5.81 ± 0.49	78.0
705	12.33 ± 2.43	6.19 ± 1.64	79.5
829	13.57 ± 0.64	7.03 ± 0.20	87.5
936	14.22 ± 0.47	7.10 ± 1.12	91.7
1264	13.06 ± 1.61	5.79 ± 0.49	84.2

## Data Availability

The data presented in this study are available on request from the corresponding author.

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
