# Peer review of "Mitigation of Galvanic Corrosion in Bolted Joint of AZ31B and Carbon Fiber-Reinforced Composite Using Polymer Insulation"

_materials, 2021, doi:10.3390/ma14071670_

Round 1

Reviewer 1 Report

Jun et al. investigated the galvanic corrosion mitigation in bolted joint of AZ31B and carbon fibre–reinforced composite using polymer insulation. I have read the manuscript and my comments are below.

  • Though the topic is very interesting, the novelty of the present study needs to be highlighted.

  • Lines 176-177: The authors stated that “The corrosion potential measured after 614 h immersion 177 were -1.26 and -1.48 VSCE for the bare and polymer-insulated joints, respectively”.  

It is clear that the corrosion potential of the insulated joints shifted towards negative direction more than that bare. Please comment.

Author Response

Jun et al. investigated the galvanic corrosion mitigation in bolted joint of AZ31B and carbon fibre–reinforced composite using polymer insulation. I have read the manuscript and my comments are below. Though the topic is very interesting, the novelty of the present study needs to be highlighted.

Lines 176-177: The authors stated that “The corrosion potential measured after 614 h immersion 177 were -1.26 and -1.48 VSCE for the bare and polymer-insulated joints, respectively”.  

It is clear that the corrosion potential of the insulated joints shifted towards negative direction more than that bare. Please comment.

[Response] The corrosion potential of AZ31B alone (w/o galvanic coupling) ranges -1.55 to -1.6 VSCE. When AZ31B is coupled with more noble materials, the corrosion potential of ‘coupled’ configuration becomes higher than that of AZ31B alone so that AZ31B experience an anodic polarization (resulting a higher corrosion rate). For bare bolt coupled AZ31B, the polarization can be approximated as: -1.55 –(-1.26) = 0.29 V. For polymer-insulated bolt with AZ31B, the polarization is approximately, -1.55-(-1.47) = 0.08 V. So, as the corrosion potential of ‘coupled’ configuration is closer to the corrosion potential of AZ31B, it can be said that the galvanic impact is lower. Thanks for the careful review.

[Action] No change was made.

Reviewer 2 Report

Dear authors,

the presented manuscript dealing with the effectiveness of a polymer insulation between metals that would usually act as a galvanic coupling is in general well drafted and written. Also the presented data (and its generation) creates a positive impressen. The presented data might be useful for design engineering, although the scientific innovation seems only average to me.

one point that seems questionable:

L146: resistance of 3.7*10-3 Mohm: this value is very low, I found it for seawater. Is this correct?

A potential application of the results should be added to increase the significance of the results.

Author Response

Dear authors,

the presented manuscript dealing with the effectiveness of a polymer insulation between metals that would usually act as a galvanic coupling is in general well drafted and written. Also the presented data (and its generation) creates a positive impressen. The presented data might be useful for design engineering, although the scientific innovation seems only average to me.

one point that seems questionable:

L146: resistance of 3.7*10-3 Mohm: this value is very low, I found it for seawater. Is this correct?

[Response] The specification in the table is for a tab water used to produce vapor and prepare 5 wt.% NaCl solution.

[Action] We revised the caption of Table A1. as: Table A1. Chemical analysis of the tap water used to prepare 5 wt.% NaCl solution and produce steam for salt spray exposure test in this work (29, 30)

A potential application of the results should be added to increase the significance of the results.

[Response] Thanks for the suggestion.

[Action] We have added a sentence in the discussion section as: The results of current work can be applied to the joint design for galvanic corrosion mitigation in lightweight multi-materials vehicles and other transportation industries, requiring fuel efficient improvement.

Reviewer 3 Report

I have now reviewed the manuscript titled “Mitigation of Galvanic Corrosion in Bolted Joint of AZ31B and Carbon Fiber–Reinforced Composite Using Polymer Insulation”. The work is appropriate for the Journal. This is a good summary of results got on the topic mentioned in the title. The work is written in appropriate style. I enjoyed reading the manuscript. Only some of the explanation regarding the results are lacking and some aspects of the paper should be improved before publication:

  1. What is the thickness of epoxy resin layer? Does coating thickness affect experimental result?
  2. A schematic of galvanic corrosion path for bare and polymer-insulated is suggested.
  3. (Figure 4) why 64, 194, 314,404 and 623 hours were selected?
  4. The addition of Load-deformation curves is suggested.

Author Response

I have now reviewed the manuscript titled “Mitigation of Galvanic Corrosion in Bolted Joint of AZ31B and Carbon Fiber–Reinforced Composite Using Polymer Insulation”. The work is appropriate for the Journal. This is a good summary of results got on the topic mentioned in the title. The work is written in appropriate style. I enjoyed reading the manuscript. Only some of the explanation regarding the results are lacking and some aspects of the paper should be improved before publication:
1. What is the thickness of epoxy resin layer? Does coating thickness affect experimental result?

[Response] As found in Figure 11c and an image below, the thickest section was approximately 200 µm, and the layer thickness was greater than 100 µm in most sections. Considering the common epoxy primer thickness, i.e. 20~40 µm for Al alloys, the coated epoxy layer in this work was sufficiently thick, so we think that there was no detrimental effect due to the thin coating thickness.

[Action] We revised a part of sentence in the result section as: revealing that the pre-coated, porous, up to 200 µm thick epoxy layer adhered firmly on AZ31B. All revised sections are presented by red fonts.

2. A schematic of galvanic corrosion path for bare and polymer-insulated is suggested. 

[Response] We agree with the reviewer’s suggestion.

[Action] We revised figure 2b and added two sentences in the experimental section: The most probable galvanic paths in a bare bolted joint are described in Figure 2b. The primary purpose of polymer insulation is to prevent/inhibit the formation of such galvanic paths.

3. (Figure 4) why 64, 194, 314, 404 and 623 hours were selected?  

[Response] We have experienced limited lab access due to COVID, so we could not meet the intended exposure times, 50 h to 600 h. However, both bare and insulated joints were exposed for the same durations.

[Action] No change was made.

4. The addition of Load-deformation curves is suggested. 

[Response] We appreciate the reviewer’s suggestion.

[Action] We added 2 ‘load vs. displacement’ figures as figs 9a and 9b in the main text. We have also added few more sentences in result section as: Figure 9a and 9b depicts representative load and displacement curves from lap shear testing for bare and polymer-insulated bolted CFRC-AZ31B joints with different corrosion exposure time. The bare joints showed greater reduction of failure load and elongation at failure with increasing exposure time. In contrast, the failure loads and displacements for polymer-insulated joints were not decreased significantly in all cases, even after 1264 hours exposure. Figure 9c and 9d present the averaged lap shear peak failure load for the bare and polymer-insulated bolted joints with increasing ASTM B117 salt spray exposure times.

Reviewer 4 Report

“Mitigation of Galvanic Corrosion in Bolted Joint of AZ31B and Carbon Fiber–Reinforced Composite Using Polymer Insulation” by J. Jun et al. is an interesting manuscript, which describe the use of polymer insulation to mitigate galvanic corrosion of bolted joints of AZ31B Mg alloy and carbon fiber–reinforced composite. The corrosion of bolted joints with and without polymer insulation was investigated by testing their exposure behavior to solution immersion and salt spray. For that, the corrosion depths and volumes were determined for joint specimens after the tests. Authors found that polymer-insulated bolted joints exhibit much lower corrosion depths and volumes and a relatively small reduction of joint strength of about 10% due to the effective mitigation of galvanic corrosion. By comparison, an important reduction of joint strength of up to 90% was observed in the group with no insulation. Moreover, cross-sectional characterization of post-corrosion joints with polymer insulation revealed local pits developed on AZ31B under galvanic influence. This indicates that a limited galvanic attack is still occur during a long salt spray exposure.

This manuscript is well-motivated and interesting, the results presented here support the view of authors, and in my opinion is suitable for publication in Materials.

However, I would like to ask authors to indicate the X-ray characteristic lines used to record the EDXS elemental maps shown on Fig. 11 e) and Fig. 12 in the manuscript.

Author Response

However, I would like to ask authors to indicate the X-ray characteristic lines used to record the EDXS elemental maps shown on Fig. 11 e) and Fig. 12 in the manuscript.

[Response] Thanks for the reviewer’s suggestion. It was K-alpha characteristic lines of C, O, F, Mg and Fe used to record the elemental maps of EDS analysis.

[Action] We have added a sentence ‘All EDS elemental maps were recorded using K-alpha characteristic X-ray of each element.’ in Fig. 11 & 12.

Round 2

Reviewer 2 Report

Dear Authors,

the changes and additions improved the manuscript significantly, I now suggest to accept it in the present form.

Kind regards